# Detection of Racist Language in French Tweets

**Natalia Vanetik** *,† and **Elisheva Mimoun** †

Software Engineering Department, Shamoon College of Engineering, 56 Bialik St., Be'er Sheva 8410802, Israel;
elishto@ac.sce.ac.il
* Correspondence: natalyav@sce.ac.il
† These authors contributed equally to this work.

**Abstract:** Toxic online content has become a major issue in recent years due to the exponential increase in the use of the internet. In France, there has been a significant increase in hate speech against migrant and Muslim communities following events such as Great Britain's exit from the EU, the Charlie Hebdo attacks, and the Bataclan attacks. Therefore, the automated detection of offensive language and racism is in high demand, and it is a serious challenge. Unfortunately, there are fewer datasets annotated for racist speech than for general hate speech available, especially for French. This paper attempts to breach this gap by (1) proposing and evaluating a new dataset intended for automated racist speech detection in French; (2) performing a case study with multiple supervised models and text representations for the task of racist language detection in French; and (3) performing cross-lingual experiments.

**Keywords:** hate speech; racist speech detection; French social media

## 1. Introduction

The exponential growth of social media such as Twitter and community forums has revolutionized communication and content publishing, but it is also increasingly being exploited for the spread of hate speech and the organization of hate activity. The term "hate speech" has been defined as "any communication that denigrates a person or group based on certain characteristics (called types of hate or classes of hate) such as race, color, ethnicity, gender, sexual orientation, nationality, religion or other characteristics" [1]. An official EU definition of hate speech [2] states that "it is based on the unjustified assumption that a person or a group of persons are superior to others; it incites acts of violence or discrimination, thus undermining respect for minority groups and damaging social cohesion."

Hate content on the internet can create fear, anxiety and threats to the safety of individuals. In the case of a business or online platform, the business or platform may lose its reputation or the reputation of its product. Failure to moderate such content can cost the company in multiple ways: loss of users, a drop in stock value, sanctions from legal authorities, etc. A news article [3] and several academic studies [4,5] indicate that during the recent COVID-19 crisis, there was a drastic increase in hate speech against people from China and other Asian countries on Twitter.

In many countries, online hate speech is a crime and is punishable by law. In this case, social media are held liable if they do not remove hateful content quickly. However, the anonymity and mobility that these media offer means that the creation and dissemination of hate speech—which can lead to hate crimes—occurs effortlessly in a virtual landscape that eludes traditional law enforcement. Manual analysis of this content and its moderation is impossible due to the enormous amount of data circulating on the internet. An effective solution to this problem would be to automatically detect and moderate hate speech comments.

In the EU, surveys and reports focusing on young people in the European Economic Area (EEA) region show an increase in hate speech and related crimes based on religious beliefs, ethnicity, sexual orientation or gender, as 80% of the respondents had experienced online hate speech and 40% had felt attacked or threatened [6,7]. The statistics also show that in the United States, hate speech and hate crimes have been on the rise since Trump's election [8].

There are many works on automatic hate speech detection (explored below in Section 2), but most of them address hate speech in English, with much fewer works in French. Additionally, racial profiling is less investigated than hate speech, especially in French. Racism is also harder to detect because often it is conveyed implicitly with stereotypes, as shown in [9,10]. To breach this gap, we have collected, annotated and evaluated a new dataset for racist speech detection in French. Our contribution is multifold: (1) we introduce a new dataset for racist speech detection in French called FTR (French Twitter Racist speech dataset); (2) we evaluate this dataset with multiple supervised models and text representations for the task of racist language detection; (3) we perform experiments for extending the FTR dataset with general hate speech data in French; and (4) we perform cross-lingual evaluations of the explored representations and methods. Our dataset is derived from Twitter and is suitable for facilitating the detection of racism on Twitter. The cross-lingual experiments are motivated by a lack of resources for racist speech detection in general, and in French in particular. In the case of successful transfer learning, one may use annotated sets in English for training systems aimed at the analysis of French texts.

The paper is organized as follows. Section 2 describes existing methods and datasets relevant to our goal. Section 3 describes the dataset and text representation and models used for its evaluation. Section 5 contains the results of experimental evaluation. Section 6 analyzes the evaluation results, and Section 7 summarizes our findings.

## 2. Background

Automatic detection of hate speech is a challenging problem in the field of natural language processing. The proposed approaches for automatic hate speech detection are based on representing the text in numerical form and using classification models on these numerical representations. In the state of the art in this domain, lexical features such as word and character n-grams [11], term frequency-inverse document frequency (tf-idf), bag of words (BoW), polar intensity and noun patterns [12] are used as input features.

Recently, word embeddings have been used as an alternative to these lexical features. A multi-feature-based approach combining various lexicons and semantic-based features is presented by Almatarneh in [13]. Liu used fuzzy methods in [14] to classify ambiguous instances of hate speech. The notion of word embeddings is based on the idea that semantically and syntactically similar words should be close to each other in an n-dimensional space [15]. The embeddings trained on a huge corpus of data capture the generic semantics of words. Word2Vec embeddings and CNN input n-character features were compared by Gambäck in [16]. Djuric [17] proposed a low-dimensional sentence representation using paragraph vector embeddings [18].

Deep learning techniques are very powerful in classifying hate speech [19]. The performance of deep learning-based approaches surpassed that of classical machine learning techniques such as support vector machines, gradient boosting decision trees and logistic regression [20]. Among the deep learning-based classifiers, a convolutional neural network captures local patterns in the text. The deep learning-based LSTM [21] model captures long-range dependencies. Such properties are important for modeling hate speech [22]. Park [23] designed a hybrid CNN by combining the word CNN and character CNN to classify hate speech. Zhang [24] designed convolutional recurrent neural networks by passing CNN inputs to GRU for hate speech classification. Del Vigna [25] showed that LSTMs performed better than SVMs for hate speech detection on Facebook. Founta [26] used an attention layer with the recurrent neural network to improve the performance of hate speech classification over a longer text sequence.

The clear majority of the offensive detection studies deal with English, partially because most available annotated datasets contain English data. For example, SemEval-2019 Task 6: Identifying and Categorizing Offensive Language in Social Media (OffensEval) was based on the Offensive Language Identification Dataset (OLID), which contains over 14,000 English tweets. The main findings of this task can be found in [27]. SemEval-2019 Task 5: Shared Task on Multilingual Detection of Hate [28] focused on detecting hate speech against immigrants and women (Task A) and detecting aggressive behavior in English and Spanish tweets (Task B). SemEval-2020 task 12: Multilingual Offensive Language Identification in Social Media [29] offered three subtasks related to offensive language detection, categorization and target identification.

Since social media became the most popular multilingual communication tool worldwide, many researchers contributed to this area by developing multilingual methodologies and annotated corpora in multiple languages. For example, Arabic [19], Dutch [30], French [31], Turkish [32], Danish [33], Greek [34], Italian [35], Portuguese [36], Slovene [37] and Dravidian [38] languages were explored for the task of general offensive speech detection.

However, there are much fewer corpora dedicated to the study of racist speech, and even fewer of them are in French. The Hate Speech Dataset Catalogue [39] contains two datasets in French only, COunter NArratives through Nichesourcing (CONAN [40]) and the Multilingual and Multi-Aspect Hate Speech Analysis dataset (MLMA [41]). The CONAN dataset is multilingual, and its French part contains 6840 comments, all of which are labeled Islamophobic. Therefore, it cannot be of help in detecting general racist content. The MLMA dataset contains 4014 comments, all of which are hate speech, with multi-class labels. Motivated by this shortage, we introduced our dataset containing annotated tweets written in French.

## 3. The FTR Dataset

Here, we present a new dataset for racist speech detection in French, titled FTR (French Twitter Racist speech dataset). In our case, we needed to retrieve many tweets that include racial speech tweets. We also needed many examples of non-racist speech tweets that contain confusing words. All the tweets were collected and annotated manually, one by one.

The data were obtained by archiving a real-time Twitter stream. The language was chosen to be French during the streaming process. The label 0 was attributed to a no racial speech tweet and 1 to a racist speech tweet. To collect tweets, we connected to the Twitter API because Twitter allows us to exploit its platform in real time or access historical tweets. We used the Python SDK frontend of the the Twitter API [42] with a list of racist terms used for filtering. Thus, the tweets have no specific format, but they were collected using the controlled vocabulary of racist speech keywords, in the desired field. The list of these terms, together with their English translation, is given in Table 1.

**Table 1.** French racist term list.

| French Expression | English Translation |
| --- | --- |
| Sal noir | Dirty black |
| Sal juif | Dirty Jew |
| Sal arabe/Sal reubeu | Dirty Arab/dirty Arab (in slang) |
| Noich | Chinese in slang |
| Bougnoul | Pejorative expression in French to designate an Arab |
| Fatma | Pejorative expression in French to designate a Muslim woman |
| Youpin/youpine | Pejorative expression in French to designate a Jew |
| Negro | Black person |
| Bamboula | Ethnic slur directed at black people |
| Boucaque | All-purpose racist slur |
| Toubab | Pejorative expression in French to designate a white |
| Niakoué | Pejorative expression in French to designate a Vietnamese |

**Table 1.** *Cont.*

| French Expression | English Translation |
| --- | --- |
| Bridé | Chinese |
| Niaqué | Chinese |
| Niaquoué | Pejorative expression in French to designate a Vietnamese |
| Sal renoi | Dirty back (in slang) |
| Manouch | Pejorative expression in French to designate a Romanian |
| Beur | Pejorative expression in French to designate an Arab |
| Sal peuple | Dirty people |
| Sale race/sal race | Dirty race |
| Nazi | Nazi |
| Crouillat | Pejorative expression in French to designate an Arab |
| Negre | Black person |
| Sal nègre | Dirty negro |
| Sal black | Dirty black |
| Bouzin | Pejorative expression in French to designate a countryman |
| Sal metisse | Dirty mixed |

In total, we have collected 2856 tweets—1929 non-racist tweets (68%), and 927 racist tweets (32%). The average number of words in a tweet (before cleaning) is 23.45, and the average number of characters is 125.15. A snippet of the dataset containing original unprocessed tweets is given in Table 2.

**Table 2.** A snippet of the FTR dattaset

| Comment | Label |
| --- | --- |
| C'est marrant comme le militantisme est toujours bridé par les intérêts personnels | 0 |
| Oui le concours du Super Nazi … Ya du monde sur la ligne de départ. | 0 |
| Je crève de faim mais je cherche une vidéo YouTube avant de manger | 0 |
| En fait Zemmour est le seul Juif que l'on peut qualifier de nazi sans encourir la désapprobation nationale. | 1 |
| Il a pris un nègre pour écrire son bouquin ? | 1 |

*Data Cleanup*

The FTR dataset was annotated by two French native speakers; the Kappa agreement coefficient between them was 0.66, which is considered to be good. In the case of disagreement, a third annotator assigned the final label. To clean the texts, we filtered out numbers, URLs and usernames.

To represent tweets as tf-idf vectors, we removed stop words. Stop words are a set of commonly used words in a language that do not help us to determine the context or the true meaning of a sentence. They can safely be ignored without sacrificing the meaning of the sentence and without any negative consequences for the final tf-idf representation. Stop words in French were derived from the NLTK FrenchStemmer SW package [43].

## 4. Dataset Evaluation

Our pipeline is based on a purely supervised approach, where every text is classified into one of two classes, based on a trained model. Models are trained on texts (training data) written in French (or English for the cross-lingual experiments), collected and annotated as described in Section 3.

We experiment with different text representations—simple tf-idf and n-grams, and semantic representations as BERT vectors. We try to verify the intuitive assumption that offensive content is easier to classify with the simple models because models generating semantic representations are not trained on data that contain a significant amount of racism examples. The pipeline for a monolingual evaluation setting is depicted in Figure 1.

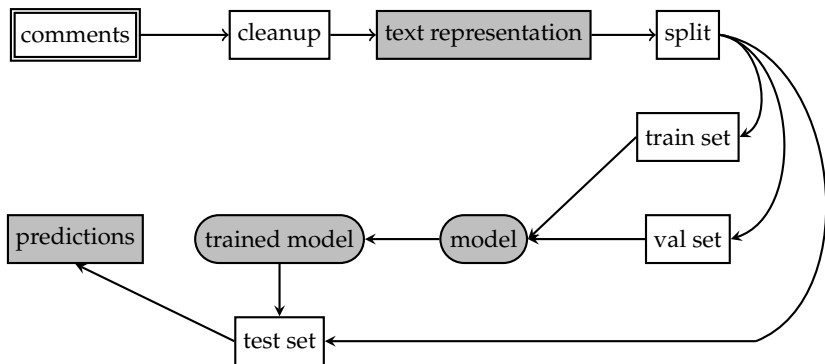

**Figure 1.** Classification pipeline for monolingual setting.

We also attempt to show that transfer learning with a different language (English) would not yield better results in comparison to the single-language (French) setting. We do hypothesize that using the same language dataset of hate speech can increase the classification accuracy, even if the dataset is not dedicated solely to racist speech. By doing so, we hope to compensate for the lack of resources on racist speech, especially in French. The pipeline for the transfer learning setting is depicted in Figure 2.

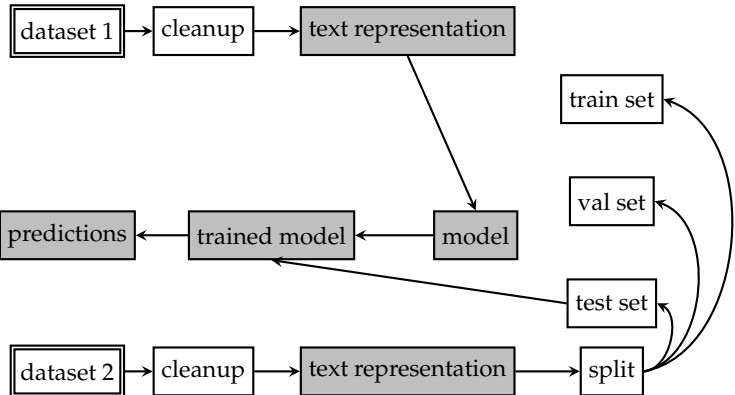

**Figure 2.** Classification pipeline for the cross-lingual and cross-domain setting.

In the case of a multilingual or multi-domain setting, the model is trained on the first dataset and on the training/validation parts of the second dataset.

### 4.1. Data

In our experiments, we use the new FTR dataset (described in Section 3), the French part of the MLMA dataset [41] and the OLID dataset [44].

The MLMA dataset is a multilingual multi-aspect hate speech analysis dataset containing Twitter posts in several languages. This dataset is used to test the multilingual multitask hate speech detection methods. We extracted the French part of this dataset and changed the annotation to fit our objectives—labels related to racism were set to 1, and the other types of hate speech received a label of 0. We selected posts that targeted a group of people according to their descent and labeled them as racist speech, and the rest were labeled as non-racism (see Table 3). It is worth noting that the MLMA dataset does not contain benign samples, i.e., texts that do not contain hate speech at all.

**Table 3.** Group labels used in the MLMA dataset.

| Group | Racism | Group | Racism |
|---|---|---|---|
| other | no | left_wing_people | no |
| individual | no | asians | yes |
| women | no | gay | no |
| african_descent | yes | jews | yes |
| immigrants | no | muslims | yes |
| arabs | yes | christian | no |
| indian/hindu | yes | hispanics | yes |
| special_needs | no | refugees | no |

The Offensive Language Identification Dataset (OLID) contains over 14,000 English tweets [27]. The tweets are annotated as offensive or not, and a type of hate speech is also given. However, there is no specific label in this dataset that indicates racism. In our experiments, we use the training part of this dataset.

Table 4 shows the basic parameters of these datasets—language, size (number of comments) and the number of positive and negative posts (denoted by pos and neg). The table also contains majority values (majority equals the ratio of the majority class in a dataset), the average, the minimal and the maximal number of words in a comment. We also specify the source of comments.

**Table 4.** Dataset statistics.

| Dataset | Lang | Size | pos | neg | Majority | avg Words | min Words | max Words | Source |
|---|---|---|---|---|---|---|---|---|---|
| FTR | Fr | 2856 | 927 | 1929 | 0.675 | 19 | 1 | 58 | Twitter |
| MLMA | Fr | 4014 | 1222 | 2792 | 0.69 | 13.3 | 1 | 27 | Twitter |
| OLID | En | 13240 | 4400 | 8840 | 0.668 | 19.1 | 1 | 61 | Twitter |

*4.2. Text Representation and Classification Models*

Our approach to text representation and classification consists of the following steps:

1. Representing texts with one of the following:
   - tf*idf vectors, where every comment is treated as a separate document;
   - word n-grams with $n = 1, 2, 3$;
   - pre-trained BERT vectors obtained from a multilingual BERT model [45].
2. Training and application of six ML supervised models (see Section 5.1) on all of the above data representations.

4.2.1. Text Representation

All data representations were computed after text cleaning as described in Section 3.

**Tf-idf**, short for term frequency-inverse document frequency, is a numerical statistic that is intended to reflect how important a word is to a document in a collection or corpus. The tf-idf value increases proportionally to the number of times a word appears in the document and is offset by the number of documents in the corpus that contain the word, which helps to adjust the model. For a term $t$ in a document $d$, the weight $W_d(t)$ of term $t$ in document d is computed as $W_d(t) = tf(t) \log(N/df(t))$, where $tf(t)$ is the frequency of term $t$ in $d$, $N$ is the size of a corpus, and $df(t)$ is the number of documents in the corpus that $t$ appears in. In our case, we treated every tweet as a separate document and the whole dataset as a corpus.

**N-grams** are the sequences of $n$ consecutive words seen in the text, where $n$ is a parameter. In our evaluation, we used the values $n = 1, 2, 3$.

**BERT sentence embeddings** of length 512 were obtained using the pre-trained multilingual distill-BERT model [45]. This multilingual BERT model is trained in 104 languages, including English and French.

Tf-idf vectors and n-grams were constructed with the help of the sklearn SW package [46]. We used the sizes $n = 1, 2, 3$ for n-grams, denoted ng1, ng2 and ng3, respectively. The values of $n > 3$ have produced inferior scores, and therefore we did not increase $n$ further.) Vector sizes for the datasets used in our experiments (FTR, MLMA and OLID) are given in Table 5. Larger vectors require more processing time during classification; however, in our experiments, this effect was not significant.

**Table 5.** Sizes of tf-idf and n-gram vector sizes.

| Representation | Dataset | Vector Size |
| --- | --- | --- |
| tf-idf | FTR | 8501 |
| tf-idf | MLMA | 8555 |
| tf-idf | OLID | 18,909 |
| ng1 | FTR | 8612 |
| ng1 | MLMA | 8655 |
| ng1 | OLID | 18,946 |
| ng2 | FTR | 30,807 |
| ng2 | MLMA | 29,601 |
| ng2 | OLID | 120,833 |
| ng3 | FTR | 40,581 |
| ng3 | MLMA | 38,793 |
| ng3 | OLID | 187,926 |
| BERT | FTR | 512 |
| BERT | MLMA | 512 |
| BERT | OLID | 512 |

### 4.2.2. Classification Models

A random forest (RF) [47] is a meta estimator that employs averaging to increase the predicted accuracy and control over-fitting by fitting several decision tree classifiers on various sub-samples of the dataset. The random forest method is an ensemble learning algorithm, which means that it is made up of several basic machine learning algorithms. Each basic machine algorithm votes for a class to forecast, and once all the basic algorithms have voted, the ensemble algorithm predicts the class with the most votes.

Logistic regression (LR) [48] is a classification algorithm that is used where the response variable is categorical. The idea of logistic regression is to find a relationship between features and the probability of a particular outcome.

Extreme gradient boosting (XGB) [49] is a special case of a boosting algorithm where errors are minimized by a gradient descent algorithm and a model is produced in the form of weak prediction models, e.g., decision trees. Gradient boosting adjusts the weights by using the gradient, using an algorithm called gradient descent, which iteratively optimizes the model loss by updating the weights. Gradient boosting uses additive modeling, in which a new decision tree is added one at a time to a model, which minimizes loss using gradient descent. The output of the new tree is combined with the output of existing trees until the loss is minimized below a threshold, or specified limit of trees is reached.

As a baseline, we used a BERT transformer [50] trained for sentence classification. We used two different models for this purpose. The first one is a multilingual model called bert-base-multilingual-cased, introduced in [50]. The second model was dehatebert-mono-french, introduced in [51]. Both of these models were fine-tuned on the train portion of our data.

### 5. Experiments

Our experiments aimed at (1) the evaluation of and comparison of various models and text representations in French; (2) cross-lingual experiments whose purpose was to understand whether transfer learning is efficient with the proposed methodology in

the case of low resources in one language and (3) multilingual experiments that studied whether adding another language such as English to a training set improves the classification accuracy.

### 5.1. Classifiers

We used random forest (RF) [52,53], logistic regression (LR) [54] and XGBoost (XGB) [55]. Fine-tuned BERT multilingual transformer [50] was used as a baseline. Two pre-trained BERT models were evaluated—one is a general multilingual model trained for sentence classification (bert-base-multilingual-cased, introduced in [50]), and the other is a French model specifically trained for hate speech detection (dehatebert-mono-french from [51]).

### 5.2. Data Setup

For the experiments on the standalone FTR dataset, RF, LR and XGB were trained on 80% of the data and evaluated on the remaining 20%. Fine-tuned BERT was trained on 75% of the data with the validation set containing 5% of the data, and it was tested on the remaining 20%. Fine-tuning was run for 10 epochs with batch size 16.

For the cross-lingual and multilingual experiments, we used representations suitable for this purpose, namely n-grams, tf-idf vectors, multilingual BERT sentence embeddings and fine-tuned BERT with pre-trained multilingual model bert-base-multilingual-cased.

In a cross-lingual setting, a model was trained on an English dataset, OLID [56], and evaluated on the test portion of the FTR dataset.

In a multilingual setting, models were trained on the OLID data and the training portion of the FTR dataset (and its validation part, in case of fine-tuned BERT), and evaluated on the test portion of the FTR dataset.

### 5.3. Software

All traditional ML models are implemented in the sklearn [46] python package. Experiments were performed on Google Colab [57] with standard settings and GPU runtime type. NumPy and Pandas libraries were used for data manipulation.

### 5.4. Evaluation Results

5.4.1. Monolingual Results

Table 6 shows the evaluation results (accuracy and F1 scores) for the standalone tests on the FTR dataset.

As can be seen, the BERT sentence embeddings text representation used with the LR classifier is the best model for the FTR in terms of accuracy and F1 measure. However, it is not better than the second-best model (1-grams with the LR classifier) in a statistically significant way, as was shown by the pairwise two-tailed Wilcoxon test [58] that produced $p$-value = 0.249. The third-best model was 1-grams with the RF classifier, and it is different from the second-best model in a statistically significant way ($p$-value $< 0.001$).

We also notice that increasing the size of an n-gram reduces both accuracy and F1 for all classification models, making it unnecessary to increase $n$ further. This shows that classification based on co-occurring word tuples of size $n > 1$ is not a good fit for the task of racism detection. However, surprisingly, the scores for $n = 1$ are on par with the ones achieved with the tf-idf representation.

**Table 6.** FTR standalone evaluation results (accuracy and F1).

| Representation | Model | F1 | Acc |
|---|---|---|---|
| bert-base-multilingual-cased | Fine-tuned BERT | 0.6473 | 0.6608 |
| dehatebert-mono-french | Fine-tuned BERT | 0.6736 | 0.7115 |

**Table 6.** *Cont.*

| Representation | Model | F1 | Acc |
|---|---|---|---|
| tf-idf | RF | 0.7018 | 0.7692 |
| tf-idf | LR | 0.6638 | 0.7587 |
| tf-idf | XGB | 0.7011 | 0.7605 |
| ng1 | RF | 0.7158 | 0.7815 |
| ng1 | LR | 0.7472 | 0.7850 |
| ng1 | XGB | 0.7158 | 0.7692 |
| ng2 | RF | 0.6409 | 0.7430 |
| ng2 | LR | 0.6382 | 0.7465 |
| ng2 | XGB | 0.5676 | 0.7133 |
| ng3 | RF | 0.6424 | 0.7448 |
| ng3 | LR | 0.6382 | 0.7465 |
| ng3 | XGB | 0.5624 | 0.7098 |
| bert | RF | 0.6690 | 0.7517 |
| bert | LR | **0.7573** | **0.7972** |
| bert | XGB | 0.7177 | 0.7657 |

### 5.4.2. Monolingual Cross- and Multi- Results

In the first experiment, we use the MLMA data [41] together with the FRT data for training. The MLMA dataset contains no benign samples; it does, however, contain various classes of hate speech. The training set in this experiment is the re-labeled MLMA dataset, and the test set is identical to the one used in standalone FTR experiments (i.e., 25% of the FTR data).

The results of this experiment for all the baselines and models are given in Table 7. Comparison to accuracy scores on standalone FTR is denoted by arrows (↓ for the lower score and ↑ for the higher score). We can see that using the MLMA alone for training does not improve the results for any of the models and text representations, and these differences are statistically significant. Therefore, we conclude that training on hate speech alone is not beneficial for our task.

**Table 7.** Cross-domain evaluation results (accuracy and F1).

| Representation | Model | Train→Test | F1 | Acc |
|---|---|---|---|---|
| bert-base-multilingual-cased | Fine-tuned BERT | MLMA→FTR | 0.5608 | 0.6294↓ |
| dehatebert-mono-french | Fine-tuned BERT | MLMA→FTR | 0.6102 | 0.6958↓ |
| tf-idf | RF | MLMA→FTR | 0.4659 | 0.6906↓ |
| tf-idf | LR | MLMA→FTR | 0.4344 | 0.6801↓ |
| tf-idf | XGB | MLMA→FTR | 0.4589 | 0.6853↓ |
| ng1 | RF | MLMA→FTR | 0.4547 | 0.6853↓ |
| ng1 | LR | MLMA→FTR | 0.5443 | 0.6923↓ |
| ng1 | XGB | MLMA→FTR | 0.4460 | 0.6766↓ |
| ng2 | RF | MLMA→FTR | 0.4080 | 0.6748↓ |
| ng2 | LR | MLMA→FTR | 0.4096 | 0.6661↓ |
| ng2 | XGB | MLMA→FTR | 0.4023 | 0.6731↓ |
| ng3 | RF | MLMA→FTR | 0.4080 | 0.6748↓ |
| ng3 | LR | MLMA→FTR | 0.4096 | 0.6661↓ |
| ng3 | XGB | MLMA→FTR | 0.4023 | 0.6731↓ |
| bert | RF | MLMA→FTR | 0.4555 | 0.6871↓ |
| bert | LR | MLMA→FTR | 0.4931 | 0.6906↓ |
| bert | XGB | MLMA→FTR | 0.4991 | 0.6888↓ |

For the second experiment, we added the 1222 racist speech posts from the MLMA dataset to the training part of the FTR dataset, in order to make it more balanced. Evaluation results for these experiments are given in Table 8. In all the cases, excluding the fine-tuned BERT trained on French hate speech data, the accuracy achieved on the enhanced data is slightly lower than on the standalone FTR (marked by ↓). The pairwise two-tailed Wilcoxon test [58] with *p*-value = 0.56 demonstrates that this result is not different in a statistically significant way from the best result achieved in the mono-domain test. The F1 score, however, is lower in all cases.

**Table 8.** Results of multi-domain evaluation (accuracy and F1).

| Representation | Model | Train→Test | F1 | Acc |
|---|---|---|---|---|
| bert-base-multilingual-cased | Fine-tuned BERT | MLMA+FTR→FTR | 0.6018 | 0.6066↓ |
| dehatebert-mono-french | Fine-tuned BERT | MLMA+FTR→FTR | 0.6487 | 0.6783↓ |
| tf-idf | RF | MLMA+FTR →FTR | 0.7063 | 0.7675↓ |
| tf-idf | LR | MLMA+FTR→FTR | 0.6643 | 0.7517↓ |
| tf-idf | XGB | MLMA+FTR→FTR | 0.6880 | 0.7570↓ |
| ng1 | RF | MLMA+FTR→FTR | 0.7050 | 0.7727 |
| ng1 | LR | MLMA+FTR→FTR | 0.7100 | 0.7552 |
| ng1 | XGB | MLMA+FTR→FTR | 0.6857 | 0.7448 |
| ng2 | RF | MLMA+FTR→FTR | 0.6575 | 0.7517↓ |
| ng2 | LR | MLMA+FTR→FTR | 0.6337 | 0.7413↓ |
| ng2 | XGB | MLMA+FTR→FTR | 0.5159 | 0.6941↑ |
| ng3 | RF | MLMA+FTR→FTR | 0.6413 | 0.7413 |
| ng3 | LR | MLMA+FTR→FTR | 0.6337 | 0.7413 |
| ng3 | XGB | MLMA+FTR→FTR | 0.5214 | 0.6976↑ |
| bert | RF | MLMA+FTR→FTR | 0.6096 | 0.7273↓ |
| bert | LR | MLMA+FTR→FTR | 0.7115 | 0.7657↓ |
| bert | XGB | MLMA+FTR→FTR | 0.6995 | 0.7587↓ |

The best model in this setting in terms of accuracy is the RF classifier applied to the n-gram representation with $n = 1$. The best F1 score is achieved by the LR model with BERT sentence embeddings text representation.

### 5.4.3. Cross- and Multilingual Cross-Domain Results

For cross-lingual and multilingual experiments, we can use multilingual text representations only. Therefore, we used BERT sentence embeddings with traditional classifiers (RF, LR and XGB), and also a fine-tuned BERT transformer pre-trained on a multilingual model, bert-base-multilingual-cased [50], as a baseline.

Table 9 contains the accuracy scores for the cross-lingual experiments, where the models were trained in English (the OLID dataset) and tested in French (the FTR dataset).

**Table 9.** Cross-lingual evaluation results (accuracy and F1).

| Representation | Model | Train→Test | F1 | Acc |
|---|---|---|---|---|
| bert-base-multilingual-cased | Fine-tuned BERT | OLID→FTR | 0.5809 | 0.6224↓ |
| bert | RF | OLID→FTR | 0.5526 | 0.7081↓ |
| bert | LR | OLID→FTR | 0.6943 | 0.7519↓ |
| bert | XGB | OLID→FTR | 0.6554 | 0.7379↓ |

As can be seen, the results (both accuracy and F1) are much worse than in the mono-lingual learning scenario (denoted by ↓) where the same language was used for training and testing. The difference is statistically significant in every case. Therefore, we conclude that pre-training a model on English for general hate speech detection does not help us in detecting racism in French.

Table 10 contains the scores for the multilingual experiment, where the models were trained in English (OLID dataset) and French (training part of the FTR dataset), and tested in French (the test part of the FTR dataset). We can see that the results are worse than in the monolingual tests, except for the accuracy score of the fine-tuned multilingual BERT model. This model, however, performs much worse than traditional classifiers that use BERT sentence embeddings as representations. Therefore, pre-training on mixed data in two languages (English and French) does not improve racist speech recognition in French.

**Table 10.** Multilingual evaluation results (accuracy and F1).

| Representation | Model | Train→Test | F1 | Acc |
|---|---|---|---|---|
| bert-base-multilingual-cased | Fine-tuned BERT | OLID+FTR→FTR | 0.6274 | 0.6801 |
| bert | RF | OLID+FTR →FTR | 0.5827 | 0.7133↓ |
| bert | LR | OLID+FTR→FTR | 0.6778 | 0.7308↓ |
| bert | XGB | OLID+FTR→FTR | 0.6681 | 0.7378↓ |

*5.5. Error Analysis*

A few examples of misclassified tweets with an explanation of the predictions of our best model (XGB classifier with tf-idf representation) are given in Tables 11 and 12.

**Table 11.** Error analysis—false negatives.

| Tweet in French | Translation | Analysis |
|---|---|---|
| hamdullilah non merci le negro est riche comme omar sy | no thanks the negro is rich like omar sy | The word 'negro' has become common and can be confusing. The meaning depends on the speaker's tone, and the context. |
| t es là ton mari c est un jambon beur tu sors un enfant bien mafe bien tiep cri encore | you are there your husband it is a ham butur you go out a child well mafe (mixed) well-typed shout again | The racist expression in this sentence is "jambon beur". The "ham butter" is a sandwich made of the two ingredients that give it its name, but this can be a racist expression designating an Arab. |
| encore un fachos néo nazi qui défend le crapaud de zemmour vive macron ces lui le prochain président on va tout faire | another neo nazi fascist who defends the toad of zemmour long live macron these him the next president we will do everything | Comparing a candidate running for president to a Nazi. It is necessary to be up to date on the political news to understand the context and the names of the politicians quoted. |
| coma nn ça meuf c jenre une noich ou truc com sa | no his girl is like a noich or something | The racist word in this sentence is "noich". It refers to a Chinese woman. It is a slang word that may not be known/understood by the model. |

**Table 11.** *Cont.*

| Tweet in French | Translation | Analysis |
|---|---|---|
| mais y a pas un peu trop de négro la dans le 11 du psg mdrr | but there is not a little too much negro there in the 11 of the psg | The word "negro" has become common and the outcome depends on the context of its use. |
| des gouts de beur | tastes of butter | "Beur" means butter and therefore designates an ingredient but also an Arab. |

**Table 12.** Error analysis—false positives.

| Tweet in French | Translation | Analysis |
|---|---|---|
| il a tout à fait raison ils viennent be travail pas ne font rien on les paie plus qu un mec qui creve en usine après ils vienne bosser au black partout pour une bouchée de pain et la france ce retrouve chômeur car nous coutont sois disant trop cher pour le pays inadmissible | He is completely right they do not come to work do nothing we pay them more than a guy who dies in the factory after they come to work in the black everywhere for a mouthful of bread and France finds itself unemployed because we cost be said too expensive for the country inadmissible | The term "bosser au black" has become a common expression for undeclared workers. The model may have interpreted the word "black" alone and not the term "bosser au black". |
| j aime trop le military lolita mais j ai peur de me faire traiter de nazi si j en porte ton psy il va nous voir arrivey en mode fancy il va faire ok pas besoin de rdv elle est folle | i love the military lolita but i'm afraid to be called a nazi if i wear it your shrink will see us arrive in fancy mode he'll do ok no need for an appointment she is crazy | The person expresses his feelings of being afraid of being called a Nazi. The model cannot detect the feelings and understand that it is not an intended statement but a feeling. |
| fidele elle a dit c est qui ce negro | faithful she said who is this negro | The word "negro" has become common and can be confusing. The meaning depends on the speaker's tone, and the context. |
| yen a marre des crève la dalle cbon | it is fed up with the slumbers it is good | In coarse language, "creve la dalle" (starving) means being hungry. The word "creve" is pejorative. The model may have stopped on this word without knowing the expression. |
| bon bah j ai pas le covid encore heureux j ai juste la crève quoi mdr | well, I don't have the covid and I'm glad I just have a cold | Same as the above. |
| sale con de merde t es un pouilleux j espère tu crève gros tas | you're a louse I hope you die fat | Offensive tweet; the reason for misclassification is unclear. |
| ptn ya des gens ils sont tellement con le seul faites d avoir leur présence à côté de toi te bride intellectuellement | fuck, some people are so stupid, the only fact of having their presence next to you restricts you intellectually | Offensive tweet; the reason for misclassification is unclear. |

Overall, slang words include informal short forms of words that are usually used in speech. Due to the lack of an existing dictionary, we have manually mapped a few slang



words to their original forms, such as "mdr" to "mort de rire", and "ajd" to "aujourd'hui". However, this action did not increase but even decreased the accuracy.

Another interesting observation is that using French stop words increased the accuracy only for the tf-idf text representation, while it decreased it for n-grams and BERT sentence embeddings. Therefore, in our experiments, we have kept the stop words for these text representations.

### 6. Discussion

Our results on the FTR show that semantic and syntactic representations of texts (tf-idf and BERT sentence embeddings) gain similar accuracy for racist language detection. We also observe that the accuracy achieved by the pre-trained fine-tuned BERT models, including the model that was trained for hate speech detection in French, is lower than the scores (both F1 and accuracy) for other models. This can be explained by the dataset's size and the fact that racist speech identification is a more difficult task than general hate speech detection.

Additionally, our hypothesis that transfer cross-lingual learning cannot be efficiently applied to racist speech in French in the case of low resources in one of them was confirmed by our experiments.

Our hypothesis on mixed-domain training in the same language is confirmed by the results. While training separately on a dataset that contains hate speech only (MLMA dataset) does not improve the scores, training on MLMA and FTR together preserves the accuracy for the best models.

### 7. Conclusions

This paper introduces and evaluates a new dataset, called FTR, for racist speech detection in French. We analyze the dataset with the help of different text representations and supervised learning methods for racist text detection in social media. We show that extending the FTR dataset with additional French data containing hate speech is beneficial because it results in better scores for almost all models and text representations. We also perform cross-lingual and multilingual experiments for testing a hypothesis about transfer learning. Unfortunately, given the current results, we cannot recommend using training datasets in a foreign language when an annotated dataset in another language is unavailable. The FTR dataset is publicly available and can be downloaded from GitHub at https://https://github.com/NataliaVanetik/FTR-dataset, accessed on 1 May 2022. In future work, we plan to extend the annotation to specify the exact span and target of offensive expressions found in tweets. We also intend to study the reasons for annotators' disagreements, as done in SemEval-2021 Task 12: Learning with Disagreements [59].

**Author Contributions:** Data curation, E.M.; Investigation, N.V. and E.M.; Methodology, N.V.; Software, N.V. and E.M.; Writing—original draft, N.V. and E.M. All authors have read and agreed to the published version of the manuscript.

**Funding:** This research received no external funding.

**Institutional Review Board Statement:** Not applicable.

**Informed Consent Statement:** Not applicable.

**Data Availability Statement:** The FTR dataset is available at https://github.com/NataliaVanetik/FTR-dataset, accessed on 1 May 2022.

**Conflicts of Interest:** The authors declare no conflict of interest.

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
