# Peer review of "Detection of Racist Language in French Tweets"

_information, doi:10.3390/info13070318_

Round 1
Reviewer 1 Report
Most “hate speech” datasets are about racial minorities but the authors consider that are generic because of the name of the phenomenon and the name of the datasets (see for instance the HatEval corpus (Basile et al., 2019) where one of the targets of hate speech were immigrants and many tweets actually containe racist comments). It is a good idea to propose a more refined approach to “racist hate speech” but the contribution of the authors in this sense is weak because they have not been rigorous in the annotation process. It is not possible to have a single annotator in a corpus that studies a social problem such as racism because there are different perspectives in what is or not racist speech. There is a lot of literature in the area that studies and tries to minimize the biases in the annotation because it is not the same if the annotators belong to the stigmatized minority or not. Hence the need for several annotators and to describe their characteristics, to organize teams of annotators balanced by gender or ethnicity and finally to provide the dataset with the disagreements included. The authors should annotate the corpus with a second annotator (in case of disagreement with the first one, also with a third one) a carry out the experiments again on the reannotated corpus.
The research has an important problem in the construction of the dataset (line 121) . The dataset has been annotated only by one person and at least two-person for each tweet would be necessary to calculate the inter annotators' agreement and a third one to resolve disagreements. Authors would need to reannotate the dataset following the conventions in the area. Racism, hate speech, the use of stereotypes is always a controversial issue, that is why a new paradigm of “perspectivisme” is proposed in the area. If they use only the vision of one person to annotate the corpus and only a second one when there is a doubt, we have neither a gold standard (consensus) nor do we have a dataset with disagreement (perspectivism paradigm).
In line 153, the authors do not describe how they label the racist vs not racist tweets in MLMA. They provided this information in 5.4.2 instead of in Section 4.1
The first reference used to define “hate speech” is presented as an official definition and it isn’t. The authors may include some of the official definitions from UE (https://www.coe.int/en/web/european-commission-against-racism-and-intolerance/recommendation-no.15) or remove “officially”.
Probably, a better reference, more academic, could be used in line 23 to refer to the increasing hate speech against Asian people during COVID-19.
The authors don’t include the reference to the EU surveys they use (line 33) and the USA statistic they mention.
Minor comments:
Line 3. Which exit from the EU? (Brexit?)
Line 6. Are the authors speaking about datasets in French???
Most “hate speech” datasets are about racial minorities but authors consider that are generic because of the name of the phenomenon and the name of the datasets. It is a good idea to propose a more refined approach to “racist hate speech” but we can’t consider the existence of a gap in datasets in English
As future work, we also encourage the authors to provide public the dataset with the disagreements following the new perspectivism paradigm (https://pdai.info/) which is highly relevant when we apply computational linguistics to social problems such as racist discourse.
Basile V., Bosco C., Fersini E., Nozza D., Patti V., Rangel F., Rosso P., Sanguinetti M. (2019) SemEval-2019 Task 5: Multilingual Detection of Hate Speech Against Immigrants and Women in Twitter. In: Proc. of the 13th Int. Workshop on Semantic Evaluation (SemEval-2019), co-located with the Annual Conference of the North American Chapter of the Association for Computational Linguistics: Human Language Technologies (NAACL-HLT 2019), Minneapolis, Minnesota, USA, June 6-7, pp. 54-63
Reviewer 2 Report
This is a very interesting paper on a dataset for racist language detection in French. Its structure is appropriate, with an interesting state-of-the art and appropriate presentation of the FTR dataset.
Dataset evaluation explains clearly used approaches. Experiments are deeply explained and the discussion is fair, explicitly indicating that the current results don’t confirm the interest of multi-lingual experiments.
Reviewer 3 Report
Dear Authors,
that's an interesting and relevant work on a sensitive subject in a world where hate and racism are dangerously rising.
This work needs some clarifications to be better understood and replicated in other idioms.
1. line 37/38 - needs a reference
2. line 44 - it must be clear if your research is only applied to Twitter messages or on any other social media. it must be clear what are the conditions to use your approach
3. sentence lines 135 to 138 - I did not understand - it did not make sense to me
4. why did figure 1 one dataset created the model, and the second dataset was used for training/validation/test? Diagram not clear
5. Please, provide a snippet of the dataset.
6. line 160 - it is not parameters, it is organization. Can you explain this table?
7. line 181 - why only 1, 2, 3. What your criteria was for that?
8. table 3 - can you explain why vector size is important in this work?
9. line 226-228 - is the percentages correct?
10. the results section is hard to read since the tables and their discussion/explanation are not closed in the text. Sometimes it is confusing.
11. Lines 240/241 - Which neural models? You just apply logistic regression and tree-based algorithms.
12. Line 249/250 - what is your explanation for such a result? It is related to item 7 of these comments (related to lines 271/272?).
13. In table 10 there are references for tweets 2292 and 2321. I could not find what it means or any reference in the text.
Hope you can refine this relevant work.
Author Response
"Please see the attachment.

Round 2
Reviewer 1 Report
This paper introduces and evaluates a new dataset, called FTR, for racist speech detection in French. The authors aim to make FTR dataset publicly available to foster research in this field. As was already commented in the first review, the dataset CANNOT be made available, and used for running experiments, with just one annotation. Datasets need always to be annotated by more than one annotator (and the inter annotator agreement needs to be calculated), and even more in case of a very subjective task like this one.
What stated by the authors - they don't have another annotator for annotating data in FVrench and they decided to just annotate 200 comments from the translating them into English (and therefore potentially introducing some noise) - doesn't hold. The authors can use Amazon Mechanical Turk to ask to annotate the dataset in French.
In the section on the related work (background), the authors could also mention shared tasks that have been organised on hate speech detection as HatEval at SemEval 2019, as it was already suggested in the first review (see below). Last but not least, sometimes racism is conveyed in an implicit form, not employing aggressive language, but using stereotypes for instance against immigrants (Sánchez et al., 2021). This should be also mentioned.
Moreover, as future work the authors could mention the new trend of taking into account ALL (and not just one...) the annotations of pre-aggregated version of each dataset (The Perspectivist Data Manifesto: https://pdai.info/). In fact, in this new trend the aim is to learn from disagreement among the annotators and some recent tasks started to consider this new paradigm in the evaluation of the systems considering also the pre-aggregated annotations and not only the gold standard that silent minority views: see for instance the SemEval-2021 shared task on learning with disagreement (Uma et al., 2021) or the DETESTS-2022 shared task on the detection of racial stereotypes https://detestsiberlef.wixsite.com/detests
Doubt to clarify:
line 312: tf-idf text representation, but decreased it for n-grams -> what weighting scheme was used for n-grams? Not tf-idf?
The authors should PROOF-READ the manuscript before resubmitting it in order to fix the typos, e.g.:
19: states that ’is based -> states that it’s based
53: for facilitating detection of racism in Twitter. -> for facilitating the detection of racism in Twitter.
144: not trained on data that contains -> not trained on data that contain (data is plural)
290: used bert sentence embeddings -> used BERT sentence embeddings
339: The FTR datasets are publicly available -> The FTR datasets is publicly available (one dataset)
Uma, A., Fornaciari, T., Dumitrache, A., Miller,T., Chamberlain,J.. Plank, B., Simpson, E. and Poesio, M. (2021). SemEval-2021 Task 12: Learning with Disagreements. In Proc. of the 15th Int. Workshop on Semantic Evaluation (SemEval-2021), pp. 338-347.
Reviewer 3 Report
Dear Authors,
thank you for the explanations and corrections. However, I have one critical question that it is not clear to me.
It is clear now you used BERT and words or expressions tokenization trying to classify racist speech. According to the new explanations, a token is composed of one word only (n=1) and that is my concern: if n=1 what you are doing is searching the string to find a token, if the token is there you detect a racist token. Why do you need to use RF for that?
Please, I hope I misunderstood the explanation. Tokens n=1 usually do not detect meaning, and that's the reason it is so hard to detect racism or other offensive speech. My doubt about your approach was largely discussed in PAN 2012 (not 100% sure if all ML models are available on the internet these days).
Round 3
Reviewer 1 Report
The authors took into account the majority of suggestions.
It is not clear to me if a statistical significant test was done (differences in the obtained results are often minor). If not, this should be done in the final version.
Moreover, I really believe that racism is often conveyed implicitely via stereotypes and this should be at least mentioned, and possibly some recent related work (e.g. (Sanche et al., 2021)) should be cited.
